# Factors Associated with Body Image and Self-Esteem in Mastectomized Breast Cancer Survivors

**DOI:** 10.3390/ijerph20065154

**Published:** 2023-03-15

**Authors:** Sergio Álvarez-Pardo, José Antonio De Paz, Ena Montserrat Romero-Pérez, Kora Mónica Portilla-Cueto, Mario A. Horta-Gim, Jerónimo J. González-Bernal, Jessica Fernández-Solana, Juan Mielgo-Ayuso, Adrián García-Valverde, Josefa González-Santos

**Affiliations:** 1Faculty of Health Science, Isabel I University, 09003 Burgos, Spain; 2Institute of Biomedicine (IBIOMED), University of León, 24071 León, Spain; 3Division of Biological Sciences and Health, University of Sonora, Hermosillo 83000, Mexico; 4Department of Health Sciences, University of Burgos, 09001 Burgos, Spain

**Keywords:** breast cancer, body image, self-esteem, sense of humor, age

## Abstract

Body image (BI) and self-esteem (SE) are two fundamental aspects in the evolution of breast cancer (BC), mainly due to surgery, treatment, and the patient’s conception of BI. A dissatisfaction with BI and low SE decreases the subject’s quality of life and increases the risk of recurrence and mortality by BC. The aim of this study is to find out if there is any degree of association between the sociodemographic data of the sample and their BI and SE. A cross-sectional, descriptive study was conducted with 198 women diagnosed with BC, aged 30–80 years, in Mexico. Women’s BI and SE were assessed using two questionnaires, Hopwood Body Image Scale (S-BIS) and Rosenberg Self-Esteem Scale (RSES). The results show significant differences in several items when the variable sense of humor is taken into account, indicating that women with a sense of humor report higher satisfaction with their BI and higher SE. The age also indicates a significantly better BI in women over 50 years of age, as well as the education level variable, where those women who had studied up to secondary reported higher satisfaction with their BI; the family history shows that those women without a family history report better SE. All these data are supported by stepwise regression, which shows that educational level and sense of humor are predictors of BI, and family history along with breast reconstruction and sense of humor are predictors as of SE. In conclusion, it is important to take into account the characteristics of women with BC, particularly age and sense of humor, in order to reduce the impact of the disease on their BI and SE with the help of a multidisciplinary team.

## 1. Introduction

Breast cancer (BC) is the most common cancer among women, being second cancer with the highest projected deaths in 2023 in the US behind lung cancer [1]. It has a 5-year survival rate for all stages (stages I, II, III, and IV) of 90% [2]. Each year, the incidence rate increases by around 0.5%, representing 31% of all female cancers [1]. This means that a large percentage of these women have to live with side effects that alter their body image (BI). This may include surgery, alopecia (which up to90% of patients experience) [3], breast asymmetry, scars, discoloration of nails on hands and feet, dermatitis, changes in body weight, or possible lymphedema resulting from the type of surgery [4,5].

The diagnosis and treatments, therefore, have a significant impact on those who suffer from it, mainly due to alterations they produce both physically and psychologically. Although it is true that the most advanced research in the field of oncological treatments tries to reduce these side effects as much as possible [6,7]. In fact, between 31% and 67% of breast cancer survivors (BCS) show some problems with their BI during treatment [8]. BI is defined as a complex mental construction that includes both our perception of the whole body and each of its parts, as well as its movement and limits. The subjective experience of attitudes, thoughts, feelings, and evaluations we have and feel, as well as the way we behave based on the cognitions and feelings we experience, are also included in the definition [9].

Its construction is evolutionary and can vary over time, so the moment and life context in which it is evaluated is crucial [9,10]. Discrepancies can be generated and, consequently, dissatisfaction with your BI. As a result, it can negatively affect self-esteem (SE) [11]. SE is an important variable in the construction of BI, which has been defined by Rosenberg (1965) [12] as “an attitude or feeling, positive or negative, toward oneself, based on the evaluation of one’s own characteristics and includes feelings of satisfaction,” indicating to what degree, the person feels capable, significant, successful, and valuable [13].

The BI of BCS is heavily affected during treatment due to the changes that occur in it. Unlike other types of cancer, BC is the only one where amputation is performed, which can be a source of negative emotions [14]. One of the main concerns of BCS is their BI, as the psychological impact is the most destructive aspect of BC [15], especially in the first year after the surgery [16]. The side effects of surgery and physical changes are the events that most affect the patient’s BI [17].

A deteriorated BI, combined with a fear of recurrence and the course of the disease, can affect the quality of life of BCS. This can cause general psychological distress, depression, social anxiety, sexual dysfunction, loss of femininity and charm, and finally, low SE [4,5,10,18,19]. Therefore, negative thoughts, irritability, lower life satisfaction, impulsivity, and lower survival rates can increase [20,21]. These symptoms can be largely alleviated with prevention strategies [8].

Sometimes the side effects become increasingly worse over time [22], and some BCS may not experience any improvement in their BI until after 5 years post-treatment [20]. A more positive BI is associated with better quality of life and better psychological well-being [10]. This leads to greater survival [23]. Likewise, an adequate SE has been related to less stress and better mental health [24].

Each of the surgical procedures and treatments employed presents a series of difficulties that will require greater or lesser attention depending on each patient. Other variables such as education, economic level, emotional state, or having a partner are also added. All of this will affect the SE and BI of the patient in one way or another. Therefore, it is necessary to gather as much information as possible so that each case is approached in a different and individualized way by each professional. In this way, the aim of this study was to identify the characteristics of BCS women related to BI and SE.

## 2. Materials and Methods

### 2.1. Study Design

This descriptive-cross-sectional study took place in Mexico, specifically in the state of Sonora. The sample consisted of 198 women who had undergone mastectomy as part of their BC treatment, regardless of the type of mastectomy (radical or conservative, with or without reconstruction).

### 2.2. Participants

The participants were recruited from various hospitals in the area of oncology, as well as from different cancer survivor associations, and were selected based on the following inclusion criteria: (i) having a confirmed diagnosis of BC, (ii) having undergone surgery as part of the treatment (mastectomy or breast-conserving surgery), (iii) being of legal age at the time of completing the questionnaire, and (iv) having been discharged by the surgeon. The main exclusion criteria were (i) having undergone bilateral mastectomy as part of the treatment and (ii) being pregnant at the time of evaluation.

### 2.3. Procedure

The evaluation of BI and SE was carried out through two methods. Firstly, a healthcare professional collected data such as the type of surgical intervention the patient had undergone (radical mastectomy or breast-conserving surgery), the stage of cancer (0, I, II, III, IV), whether breast reconstruction had been performed (yes or no), and the type of treatment the patient was receiving at the time of evaluation (CH, RTH, target therapy, immunotherapy, and hormone therapy). Then, participants completed a sociodemographic and health questionnaire validated by experts in the field. All of them participated in the study voluntarily and provided written informed consent after being explained the nature and purpose of the study, which was approved by the Human Research Bioethics Committee of the Department of Medicine and Health Sciences at the University of Sonora (DMCS/CBIDMCS/D-50).

### 2.4. Assessments

In the sociodemographic questionnaire, data such as current employment status (unemployed, employed, incapacitated, or retired), age organized by ranges (<50 years, between 50 and 65 years, and >65 years), level of education at the time of evaluation (basic level including primary and secondary education, the upper secondary level having completed high school, higher education having a university degree or postgraduate degree, and a level classified as “other” (including technical studies or training courses), marital status at the time of evaluation (with a partner, widowed, divorced, or single), family history of cancer (with a first-degree family history, with a second-degree family history, or without a family history), whether they were aware of the diagnosis (no or slightly aware, aware), sense of humor (without sense of humor or with sense of humor), and finally whether they received psychological treatment after surgery (yes or no) were obtained.

BI was measured using the Body Image Scale (S-BIS) [25]. The original version of the scale was developed by Hopwood, Fletcher, and Ghazal [26]. This scale consists of ten items developed to assess BI from three dimensions, affective, behavioral, and cognitive. Scores are recorded on a 4-point Likert scale ranging from 0, “not at all”, 1 “a little”, 2 “quite a bit”, 3 “very much”. Higher scores indicate greater problems related to BI [9]. The total score ranges from 0 to 30, with scores greater than 10 indicating dissatisfaction and scores equal to or less than 10 indicating satisfaction with BI [27,28]. The S-BIS scale has good reliability, internal consistency (α = 0.93) [29], and validity according to different studies that have analyzed this questionnaire [30].

In order to evaluate SE, the Rosenberg Self-Esteem Scale (RSES) was used [12]. The original version of the scale was developed by Morris Rosenberg in 1965 with the aim of exploring personal worth and self-respect. It is one of the most widely used scales globally for assessing self-esteem and has been translated and validated into Spanish [31]. The RSES consists of 10 items that refer to self-esteem and self-acceptance on a 4-point Likert scale, 1 (strongly agree), 2 (agree), 3 (disagree), 4 (strongly disagree); in this case, a minimum score of 10 points and a maximum of 40 points is possible, for which the negatively worded items (3, 5, 8, 9, 10) are first reversed and then all items are summed.

According to the sum of the items, the scale is divided into three categories: normal or high SE (30–40 points), medium SE, without serious problems but in need of improvement (26–29 points), and low SE, indicating significant problems (<26 points). This scale has a Cronbach’s alpha of 0.87, reliability of 0.74 [31], and has been validated according to different studies that have analyzed this questionnaire [32]. Both scales can be viewed in Appendix A for further information.

### 2.5. Statistical Analyzis

Data are presented as the number of cases and (% of total) or as mean ± standard deviation of the mean (SD). Statistical analyzis was performed with the software SPSS version 25 (IBM-Inc., Chicago, IL, USA).

Groups were created according to data collected at the time of evaluation. For age (<50 years vs. ≥50 years), this cut-off has been justified in the existing literature [33]. Years since diagnostic or elapsed years (<5 years vs. ≥5 years), this 5-year cut-off has been used and justified in the current literature [34]. Whether they had knowledge of the diagnosis or not, patients were asked about the information they had regarding their BC (diagnosis, treatment, and prognosis) and reflected on their perception of the information they had(none or little vs. knowledge). The other variables were classified as: level of education(up to high school vs. university or higher), type of surgery (mastectomy vs breast-conserving therapy), receiving treatment at the moment of the evaluation (without receiving treatment vs receiving treatment), marital status (with partner vs. without partner), medical history (without medical history vs. with medical history), employment status (unemployed vs. employee), sense of humor (without sense of humor vs. with sense of humor) and psychological support (without psychological support vs. with psychological support).

The differences in the total BI and SE scores between groups were tested using bivariate analyzis, performing an independent samples t-test, with the group to which each patient belonged as a fixed factor. In order to analyze factors associated with BI and SE. A Pearson correlation is also performed between the two scales used with the age and the base years. A stepwise regression was used with age, elapsed years, treatment, type of surgery, family history, marital status, employment status, level of education, reconstruction, sense of humor, knowledge of diagnosis, and psychological support as independent variables, and BI and SE scores as dependent variables. Effect sizes are presented as Cohen’s d, based on absolute differences between groups, with 0.2, 0.5, and 0.8 considered small, medium, and large effect sizes, respectively [35].

## 3. Results

The demographic and descriptive characteristics of the 198 women who participated in the study are reported in the following article [34]. In this case, more variables were taken into account. Psychological support, where 60% did not receive any type of psychological support; a sense of humor, where 90% of the sample considered themselves to have a sense of humor; and finally, knowledge of the diagnosis, where more than 60% had knowledge of it.

The analyzis of BI was carried out using the S-BIS, as shown in Table 1. The data revealed that 34 out of 198 BCS (17.17%) felt dissatisfied with their BI. A score greater than 10 on the scale was obtained (not expressed data). The groups up to secondary school (4.15 ± 4.97) and without medical history (3.87 ± 4.75) obtained the lowest score on the scale. This indicates greater satisfaction. The highest scores were in the groups without a sense of humor (7.61 ± 6.26) and from secondary school (6.07 ± 5.99), although the average score does not represent dissatisfaction with BI since it is not higher than 10 points on the S-BIS scale.

In S-BIS, a significant relationship was found in the variable age (*p* = 0.046). A score of 5.82 ± 6.20 was obtained in the group <50 years and 4.25 ± 4.76 in ≥50 years. This shows that older women report greater satisfaction with their BI. A significant relationship was also found in the variable educational level (*p* = 0.015), with a score of 4.15 ± 4.97 in the group up to secondary school and 6.07 ± 5.99 in the group from secondary school BCS with lower educational level report greater satisfaction with their BI. Finally, a significant relationship was also observed in the variable sense of humor (*p* = 0.017). A score of 7.61 ± 6.26 was obtained in BCS without a sense of humor and 4.62 ± 5.31 in BCS with a sense of humor. Consequently, patients with a higher sense of humor report a lower score on the scale and, therefore, greater satisfaction with their BI. According to Cohen, the effect sizes of these findings are small to medium in the variables of age and educational level; and medium to large in the variable sense of humor.

Additionally, a correlation between the S-BIS scale and the variables age and elapsed years is added. Showing a significant and negative correlation with age r_(198)_ = 0.148, *p* = 0.05. For the other variables analyzed, no significant correlation was found.

The analysis of SE was carried out through the RSES, as shown in Table 2. None of the BCS in the sample indicated low SE. Only 25 out of 198 (12.62%) of the BCS showed moderate SE, with scores ranging from 26 to 29 on the scale. The87.37% of the sample scored high SE with a score above 30 on the RSES (not expressed data). The variables with the lowest scores were those BCS who had a slight knowledge of the diagnosis (34.64 ± 5.82) and those without a sense of humor (32.14 ± 5.35). They do not represent average or low SE since the average score is above 30. On the other hand, the highest scores can be found in BCS without a medical history (36.89 ± 3.03) and in the reconstruction group (37.69 ± 1.84).

In RSES, a significant relationship was found in the variable family history (*p* = 0.014). With a score of 36.89 ± 3.03 in the group without medical history and 34.97 ± 5.14 in the group with a medical history, it is reported that those BCS without a family history have higher SE than those with a family history of cancer. In the sense of humor variable (*p* = 0.001), with a score of 32.14 ± 5.35 in the group without a sense of humor and 35.84 ± 4.56 in the group with a sense of humor, it is reported that BCS with a sense of humor has a higher score on the RSES, indicating higher SE. According to Cohen, the effect sizes of these findings range from small to medium in the family history variable and from medium to large in the sense of humor variable.

The stepwise regression model identified significant associations between education level and sense of humor with BI. In addition, SE with a sense of humor, family history, and reconstruction, as shown in Table 3.

In this case, the level of education is a predictor of BI. The higher level of education, the higher score on the BI scale, and therefore, the greater dissatisfaction. On the other hand, sense of humor is a predictor of BI, indicating that the greater sense of humor, the lower the score on the BI scale and the greater satisfaction with de BI.

Regarding SE, a sense of humor, family history, and reconstruction are predictors of SE. A higher sense of humor predicts higher SE, as does undergoing reconstruction. Conversely, having a family history predicts lower scores on the SE scale.

## 4. Discussion

In general, the most transcendent factors that affect BI and SE during the course of breast cancer are the type of surgery, alopecia, changes in the skin, and alterations in body weight (gain of fat mass and loss of muscle mass) [14]. The surgery performed on BC patients will have a significant impact on their BI, due to the possible amputation of one or both breasts, scarring, numbness, and swelling [14]. BCS indicates that after surgery, they face problems with poor cosmetics, loss of femininity, or even acute complications [36]. Women who undergo immediate reconstruction are the ones who are less vulnerable to experiencing a deterioration in their BI and low SE [37,38], as they are not directly exposed to the problem of asymmetry or feeling less feminine [9].

Similarly, an important variable to consider in each of these treatments is the degree of disease spread to the lymph nodes [14]. Surgical intervention can lead to significant psychological, social, and sexual morbidity that can persist in a woman’s mind for a long time [5,39]. Even 5 years after being operated on, they may still feel less attractive [40]. This is because the patient feels they have a potentially life-threatening disease [39]. Therefore, undergoing a mastectomy can be considered a potentially traumatic event; while cancer is considered a source of psychological stress that complicates regular psychosocial functioning [36].

Due to the loss of body hair caused by chemotherapy and the changes in the skin caused by radiation therapy (such as discoloration of nails, sallow complexion, skin irritation, the appearance of oozing wounds, and edema in the radiated area), they appear unhealthy. This represents one of the most prominent side effects that greatly affects the appearance of BCS [14].

Finally, changes in body composition, such as loss of muscle mass, a gain of fat mass, and metabolic syndrome resulting from treatment, lead to negative self-judgment and the perception that they did not measure up to their own ideals or standards, feeling self-conscious, sad, and uncomfortable with their body [17].

There are a wide variety of variables that influence the BI and SE of BC patients, from age to socioeconomic status. It is essential that the most vulnerable BCS in this regard can develop a new BI, discovering alternative remedies and solutions to the restrictions that come with surgery. In this way, they can better adapt to the physical and emotional aspects of their recovery. Achievement and improvement in all areas will increase their quality of life. If this cannot be achieved, it will result in significant psychological, social, and sexual morbidity since it has a very negative impact and lasts for a long period of time in the woman’s mind [5,39,41].

Our results indicate that BCS in the group ≥ 50 years old have higher satisfaction with their BI than younger BCS (*p* = 0.046) and a higher SE, although not reaching significance, with both having a small to medium size. This is in line with most studies [14,19,42,43,44] that indicate that dissatisfaction with BI and SE in younger BCS is related to the onset of early menopause triggered by treatment. In addition, younger women have lower sexual satisfaction and enjoyment as well as greater concern about their appearance, a tendency to compare themselves to their peers and to receive more aggressive treatment in a greater number of doses [19,42]. Younger BCS tend to readapt mentally worse to the crisis of having a disease such as breast cancer, as it may distort all short and medium-term plans, and the sequelae can undermine their expectations about their future lifestyle, which may have to be modified [14]. Finally, in Latin American countries, reconstruction is not included in the healthcare system. It must be paid for out of pocket, which causes greater work and economic stress for younger women compared to older women whose children will be independent and will have fewer problems paying for surgery [43]. Being at a younger age is a predictive factor of poorer BI [42], and older age is a protective factor against problems with BI and SE. This is because there is less pressure to conform to youth beauty standards and a greater sense of security and self-comfort as one age [44].

Regarding the performance of breast reconstruction surgery after mastectomy, in the stepwise regression, the reconstruction variable predicts a higher score on the SE scale. This is consistent with other studies indicating that women who undergo breast reconstruction have a lower overall level of distress because they do not experience the severe shyness that accompanies the loss. They also have a higher degree of sexual satisfaction, better relationships with their partners, and feel more attractive. Immediate reconstruction is more effective than delayed reconstruction [9,39,42,45].

On the other hand, our results indicate that patients with an education level up to secondary school have higher satisfaction with their BI (*p* = 0.015) compared to those with a higher level of education. These results are supported by stepwise regression, which shows that education level is a predictor of BI, where higher education level predicts higher scores on the BI scale and, therefore, greater dissatisfaction with BI. This is consistent with other research indicating that education level is a factor associated with concerns about BI [14,40,46,47,48]. Reasons why a lower education level is related to greater satisfaction with BI include fear associated with returning to work, especially for BCS with a higher education level, who often hold qualified and higher-responsible positions compared to those with less education. Additionally, higher education level correlates with higher socioeconomic status, which helps BCS to have more options for hiding certain treatment side effects, such as real hair wigs or high-quality cosmetics, which give them more concern and importance regarding how their BI will be perceived [14]. It should be noted that early detection of breast cancer is higher in women with high education levels, meaning that the treatment and surgery will be less aggressive than in cases where the disease is more advanced, which impacts the BI and SE of BCS [46,47,48].

Regarding the variable sense of humor, our results indicate that BCS with a sense of humor has higher satisfaction with their BI (*p* = 0.017) and a higher SE (*p* = 0.001) than BCS without a sense of humor. These findings were supported by the stepwise regression, which indicated that a sense of humor was a predictor of greater BI and SE. This is the first study that includes this variable as a conditioning factor of self-perception in BCS, although there are works that have suggested that humor and laughter are important tools for coping with cancer, both in the daily life of BCS and in their communication with healthcare professionals [46]. As well as studies indicating that positivity and therapies that increase it are capable of improving the quality of life of women with breast cancer [49].

In the variable family history, BCS without family history report higher SE (*p* = 0.014) than those with family history, with a small to medium effect size in both questionnaires. Stepwise regression confirms these results, showing that having a family history predicts a lower SE. Studies on high-performance athletes have shown that mental representation of future events or stressful visualizations negatively affects their athletic performance [50,51,52,53]. Therefore, from this point of view, it could be argued that a negative visualization of the context in BCS with family history could influence the perception of BI by anticipating undesired changes in BI and thus affecting SE.

On the other hand, having knowledge of the diagnosis and receiving correct information from healthcare professionals about the disease, its risks, complications, treatment, and prognosis is crucial. Lack of information and uncertainty about what is happening can generate distress and false expectations about future BI, directly affecting SE [5,22,54]. The perceived social support received by BCS is a very important element to take into account, and it is worth noting that those who perceived greater social support show fewer problems with their BI. This not only happens in BCS but also in healthy women, and it is closely related to having a partner since survivors who are married have greater psychological and physical support from their partners than unmarried/divorced survivors [55].

In general, the literature indicates that dissatisfaction with BI and low SE often leads to lower quality of life and an increased risk of depression and anxiety [56]. It is closely related to lower breast cancer survival rates [33]. Therefore, the following variables, age, time elapsed since diagnosis, type of surgery, knowledge of the diagnosis, sense of humor, psychological support, and family history, should be studied in depth to determine their influence on the quality of life and adverse mental health outcomes. However, the scope of this study is limited to a sample population in a single state in Mexico, and therefore, cannot be generalized to the entire population of Mexico or the world population. Moreover, the effect of culture must be taken into account when attempting to extend the results to a non-Latin population. Additionally, there is difficulty in that BI includes both objective and subjective components, as different individuals place different weights on certain areas of life [9]. There gore, the context in which SCM is situated with respect to her daily life should be studied more thoroughly. Hence, there is a need to further research on this issue, which affects so many women around the world. Addressing new cultures and healthcare systems is an important point of research.

## 5. Conclusions

The results of this study showed a significant correlation between BI and the variables of age, education level, and sense of humor, as well as between SE and family history and sense of humor. This is reinforced by the results obtained from the stepwise regression, adding the variable of breast reconstruction as a predictor of SE, among other variables that provide practical results, such as time elapsed since diagnosis and patient knowledge about their diagnosis, treatment, and prognosis of breast cancer. The importance of future research that takes into account social support and a sense of humor is highlighted, as it is essential to better understand the profiles of BCS who are at higher risk of experiencing dissatisfaction with their BI or low SE. The purpose is to develop individualized interventions based on the characteristics of each patient, such as group therapies (peer discussion groups, cognitive-behavioral therapies, and psychosocial programs), beauty and makeup treatments [57], couple’s therapy (as report communication problems, increased conflicts or even breakups during the course of the disease) [58], and finally, the prescription of physical exercise [59] to increase satisfaction with BI, SE, and cancer survival.

## Figures and Tables

**Table 1 ijerph-20-05154-t001:** Association between Characteristics and Body Image in Breast Cancer Survivors (BIS).

Variable	Group		*p*	Cohen’s d	95% (CI)
Age	<50 years (*n* = 86)	≥50 years (*n* = 112)			
	5.82 ± 6.20	4.25 ± 4.76	0.046	−0.289	−0.572 to −0.007
Elapsed years	<5 years (*n* = 145)	≥5 years (*n* = 53)			
	5.09 ± 5.59	4.50 ± 5.20	0.506	−0.107	−0.422 to 0.207
Type of surgery	Mastectomy (*n* = 72)	Breast-conserving therapy (*n* = 126)			
	4.43 ± 5.94	5.23 ± 5.20	0.325	0.146	−0.144 to 0.436
Receiving treatment	Without receiving treatment (*n* = 103)	Receiving treatment (*n* = 95)			
	4.99 ± 5.08	4.88 ± 5.91	0.892	−0.020	−0.299 to 0.259
Reconstruction	Without reconstruction (*n*= 185)	With reconstruction (*n* = 13)			
	4.91 ± 5.51	5.23 ± 5.15	0.843	0.058	−0.504 to 0.621
Family history	Without medical history (*n* = 49)	With medical history (*n* = 149)			
	3.87 ± 4.75	5.28 ± 5.67	0.118	0.258	−0.065 to 0.582
Marital status	Without partner (*n* = 46)	With partner (*n* = 152)			
	5.13 ± 6.01	4.88 ± 5.32	0.788	−0.046	−0.375 to 0.284
Employment situation	Unemployed (*n* = 141)	Employee (*n* = 57)			
	4.99 ± 5.57	4.80 ± 5.27	0.830	−0.035	−0.342 to 0.273
Level of education	Up to secondary school (n = 117)	From secondary school (n = 81)			
	4.15 ± 4.97	6.07 ± 5.99	0.015	0.355	0.069 to 0.640
Knowledge of diagnosis	No or slightly (*n* = 77)	Yes with knowledge (*n* = 121)			
	5.07 ± 5.40	4.85 ± 5.54	0.777	−0.040	−0.326 to 0.246
Sense of humor	Without sense of humor (*n* = 21)	With sense of humor (*n* = 177)			
	7.61 ± 6.26	4.62 ± 5.31	0.017	−0.552	−1.008 to −0.097
Psychological support	Without psychological support (*n* = 120)	With psychological support (*n* = 78)			
	4.80 ± 5.69	5.15 ± 5.15	0.658	0.064	−0.221 to 0.349

*p* < 0.05 significance level. *p*: *p*-value.

**Table 2 ijerph-20-05154-t002:** Association between Characteristics and Self Esteem in Breast Cancer Survivors (RSES).

Variable	Group		*p*	Cohen’s d	95% (CI)
Age	<50 years (*n* = 86)	≥50 years (*n* = 112)			
	35.19 ± 5.73	36.64 ± 3.91	0.517	0.303	0.020 to 0.586
Elapsed years	<5 years (*n* = 145)	≥5 years (*n* = 53)			
	35.20 ± 5.05	36.13 ± 3.89	0.225	0.195	−0.120 to 0.510
Type of surgery	Mastectomy (*n* = 72)	Breast-conserving therapy (*n* = 126)			
	35.61 ± 3.97	35.35 ± 5.19	0.720	−0.054	−0.344 to 0.235
Treatment	Without treatment (*n* = 103)	With treatment (*n* = 95)			
	35.01 ± 5.45	35.91 ± 3.90	0.188	0.189	−0.091 to 0.468
Reconstruction	Without reconstruction (*n*= 185)	With reconstruction (*n* = 13)			
	35.29 ± 4.88	37.69 ± 1.84	0.080	0.505	−0.059 to 1.070
Family history	Without medical history (*n* = 49)	With medical history (*n* = 149)			
	36.89 ± 3.03	34.97 ± 5.14	0.014	−0.408	−0.733 to −0.083
Marital status	Without partner (*n* = 46)	With partner (*n* = 152)			
	36.00 ± 3.27	35.28 ± 5.14	0.374	−0.151	−0.481 to 0.179
Employment situation	Unemployed (*n* = 141)	Employee (*n* = 57)			
	35.14 ± 5.06	36.19 ± 3.92	0.165	0.220	−0.088 to 0.529
Level of education	Up to secondary school (*n* = 117)	From secondary school (*n* = 81)			
	35.35 ± 5.11	35.59 ± 4.28	0.727	0.050	−0.233 to 0.333
Knowledge of diagnosis	No or slightly (*n* = 77)	Yes with knowledge (*n* = 121)			
	34.64 ± 5.82	35.95 ± 3.91	0.060	0.276	−0.011 to 0.563
Sense of humor	Without sense of humor (*n* = 21)	With sense of humor (*n* = 177)			
	32.14 ± 5.35	35.84 ± 4.56	0.001	0.796	0.337 to 1.255
Psychological support	Without psychological support (*n* = 120)	With psychological support (*n* = 78)			
	35.35 ± 4.19	35.60 ± 5.58	0.717	0.052	−0.233 to 0.337

*p* < 0.05 significance level. *p*: *p*-value.

**Table 3 ijerph-20-05154-t003:** Stepwise regression analysis between Characteristics and Body Image and Self Esteem in Breast Cancer Survivors.

	Unstandardized		Standardized Coefficients		
Model	B	Std. Error	Beta	t	Sig.
**(Constant)**	4.905	1.664		2.948	0.004
**Educational level**	1.781	0.777	0.160	2.292	0.023
**Sense of humor**	−2.769	1.241	−0.156	−2.232	0.027
Dependent variable Body image R^2^ = 0.044					
	**Unstandardized**		**Standardized Coefficients**		
**Model**	**B**	**Std. Error**	**Beta**	**t**	**Sig.**
**(Constant)**	35.581	1.689		21.072	0.000
**Sense of humor**	3.520	1.055	0.227	3.337	0.001
**Family history**	−1.970	0.757	−0.178	−2.604	0.010
**Reconstruction**	2.650	1.317	0.138	2.012	0.046
Dependent variable Self Esteem R^2^ = 0.019					

B—Regression Coefficient; Std. Error—Standard Error; Sig.—Significance; R^2^—Correlation Coefficient.

## Data Availability

Not applicable.

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
