# Peer review of "Factors Associated with Body Image and Self-Esteem in Mastectomized Breast Cancer Survivors"

_ijerph, 2023, doi:10.3390/ijerph20065154_

Round 1
Reviewer 1 Report
Dear authors,
Your manuscript, "Factors associates with body image and self-esteem in mastectomized breast cancer survivors", shows the results of the self-perception of breast cancer patients after surgery. I believe this manuscript would support the previous publication of this research group (Alvarez-Pardo et al., 2023). However, I would like to comment on some concerns.
Major comments
1. I think it is relevant to include the elapsed time from the surgery (in addition to the elapsed time from the diagnosis). It may be useful to control this variable.
2. This study has converted numeric variables into a few categories. Since S-BIS and RSES are numerical data, I suggest evaluating the correlation (in addition to categorical comparisons) with other clinical information.
Minor comments
3. Please, check for some typos across the text. For example, associates -> associated (Title).
Author Response
Mr. Sergio Álvarez Pardo
Department of health sciences
University of Burgos, Paseo Comendadores s/n.
Burgos, 09001, Spain
Tel. (+34) 947499108
Email: sergioal96@hotmail.com
04-03-2023
IJERPH. Subject: Submissions Needing Revision
Dear editor.
Thank you very much for inviting us to submit our response to reviewers for our manuscript (ijerph-2260388) entitled: “Factors associates with body image and self-esteem in mastectomized breast cancer survivors”
We have checked our manuscript according to the Academic Editor, the reviewers’ comments and the Journal requirements. We have also responded to some comments from reviewers point by point).
We would be very grateful if you could consider our manuscript to be published in your journal.
Yours sincerely,
Sergio Álvarez Pardo, Sport Science Degree
- Response to Reviewer 1:
First of all, we would like to express our sincere gratitude for all comments and suggestions received from the Reviewer 1. This information has certainly enriched the text for its best understanding, thank you very much indeed. We have clarified the reviewer1’s questions. We have introduced the required changes both in our answers to the specific comments and in the final manuscript.
Your manuscript, "Factors associates with body image and self-esteem in mastectomized breast cancer survivors", shows the results of the self-perception of breast cancer patients after surgery. I believe this manuscript would support the previous publication of this research group (Alvarez-Pardo et al., 2023). However, I would like to comment on some concerns.
Major comments
- I think it is relevant to include the elapsed time from the surgery (in addition to the elapsed time from the diagnosis). It may be useful to control this variable.
Response: Thank you for pointing this out. At this point in the research we do not have this type of information about our sample, we could not collect it at the time due to lack of information.
- This study has converted numeric variables into a few categories. Since S-BIS and RSES are numerical data, I suggest evaluating the correlation (in addition to categorical comparisons) with other clinical information.
Response: Thank you for pointing this out. We have added this information in the manuscript (See lines 178-179 and 216-218).
Minor comments
- Please, check for some typos across the text. For example, associates -> associated (Title).
Response: Thank you for pointing this out. We have amended this issue
We hope we have now answered all your comments and we are looking forward to hearing from you again.
Thank you very much,
Sergio Álvarez Pardo, Sport Science Degree

Reviewer 2 Report
Thank you for the opportunity to read this manuscript and congratulations to the authors for their work. I highly appreciate the text submitted for review.
Here are some suggestions for improvement:
· The introduction section is too long and it is better to be shorter and clear.
· The method section can be more precise for a better understanding of the discussion results.
· Please bring the questionnaire as a supplement file
· Please bring the analysis of the Reliability and Validity of the Questionnaire results.
· The study approval by the Human Research Bioethics Committee of the Department of 94 Medicine and Health Sciences at the University of Sonora (DMCS/CBIDMCS/D-50) should be placed at the last paragraph of the methodology section.
· Please present the results section in a more precise form.
Author Response
Mr. Sergio Álvarez Pardo
Department of health sciences
University of Burgos, Paseo Comendadores s/n.
Burgos, 09001, Spain
Tel. (+34) 947499108
Email: sergioal96@hotmail.com
04-03-2023
IJERPH. Subject: Submissions Needing Revision
Dear editor.
Thank you very much for inviting us to submit our response to reviewers for our manuscript (ijerph-2260388) entitled: “Factors associates with body image and self-esteem in mastectomized breast cancer survivors”
We have checked our manuscript according to the Academic Editor, the reviewers’ comments and the Journal requirements. We have also responded to some comments from reviewers point by point).
We would be very grateful if you could consider our manuscript to be published in your journal.
Yours sincerely,
Sergio Álvarez Pardo, Sport Science Degree
- Response to Reviewer 2:
First of all, we would like to express our sincere gratitude for all comments and suggestions received from the Reviewer 2. This information has certainly enriched the text for its best understanding, thank you very much indeed. We have clarified the reviewer2’s questions. We have introduced the required changes both in our answers to the specific comments and in the final manuscript V2.
Thank you for the opportunity to read this manuscript and congratulations to the authors for their work. I highly appreciate the text submitted for review.
Here are some suggestions for improvement:
- The introduction section is too long and it is better to be shorter and clear.
Response: Thank you for your comment, we have modified this issue by deleting some information that was not in line with the theme of the article, but we had to add information at the request of another reviewer.
- The method section can be more precise for a better understanding of the discussion results.
Response: Thank you for your comment, we have modified this section for better understanding.
- Please bring the questionnaire as a supplement file
Response: Thank you for your comment, we have attached the supplement file.
- Please bring the analysis of the Reliability and Validity of the Questionnaire results.
Response: Thank you very much for pointing this out, we have added this information in the manuscript (See lines 141-142 and 154-155)
- The study approval by the Human Research Bioethics Committee of the Department of 94 Medicine and Health Sciences at the University of Sonora (DMCS/CBIDMCS/D-50) should be placed at the last paragraph of the methodology section.
Response: Thank you for your comment, we have modified this section (See lines 117-121)
- Please present the results section in a more precise form
Response: Thank you for your comment, we have modified this section for better understanding
We hope we have now answered all your comments and we are looking forward to hearing from you again.
Thank you very much,
Sergio Álvarez Pardo, Sport Science Degree

Reviewer 3 Report
Comments to the Authors:
This article describes the factors that affect the body image (BI) and self-esteem (SE) of breast cancer (BC) survivors after mastectomy. Women over 50 had better BI, and education level and sense of humor were predictors of BI, while family history, breast reconstruction and sense of humor were predictors of SE. Considering the age and sense of humor and other characteristics of women with BC, this study can help reduce the impact of the disease on their BI and BE and improve their happiness, which has good research significance. This manuscript is suitable for publication in Int. J. Environ. Res. Public Health, but the following modifications are recommended prior to publication.
1. There are some formatting errors in the article, please check and modify them.
2. On page 2, line 61, "BC is the only one where mutilation is performed" is not precise, please replace it with an appropriate expression.
3. Page 2, line 70 "lower survival rates may be increased" is not fluent in language, please modify this.
4. In Tables1 and 2, "With reconstruction" and "Without sense of humor" are not very convincing due to the small number of people. It is suggested to investigate more data.
5. Please mark where Table 3 is described in the text.
6. "Age" is mentioned in the keyword, but there is not much discussion about it in the text, please add some discussion about it.
7. The format of references is not uniform. For example, Ref. 6, change “2021, Volume 13, 701–709” to “2021, 13, 701–709”.
8. Introduction, the authors talked about cancer, the latest cancer diagnosis and treatment should be mentioned. In order to support this statement, the following recently published important related papers should be cited: Adv Mater. 2022, 34, 2106388; Sci. China: Chem. 2023, 66, 10.1007/s11426-022-1477-x.
Author Response
Mr. Sergio Álvarez Pardo
Department of health sciences
University of Burgos, Paseo Comendadores s/n.
Burgos, 09001, Spain
Tel. (+34) 947499108
Email: sergioal96@hotmail.com
04-03-2023
IJERPH. Subject: Submissions Needing Revision
Dear editor.
Thank you very much for inviting us to submit our response to reviewers for our manuscript (ijerph-2260388) entitled: “Factors associates with body image and self-esteem in mastectomized breast cancer survivors”
We have checked our manuscript according to the Academic Editor, the reviewers’ comments and the Journal requirements. We have also responded to some comments from reviewers point by point).
We would be very grateful if you could consider our manuscript to be published in your journal.
Yours sincerely,
Sergio Álvarez Pardo, Sport Science Degree
- Response to Reviewer 3:
First of all, we would like to express our sincere gratitude for all comments and suggestions received from the Reviewer 3. This information has certainly enriched the text for its best understanding, thank you very much indeed. We have clarified the reviewer3’s questions. We have introduced the required changes both in our answers to the specific comments and in the final manuscript V2.
This article describes the factors that affect the body image (BI) and self-esteem (SE) of breast cancer (BC) survivors after mastectomy. Women over 50 had better BI, and education level and sense of humor were predictors of BI, while family history, breast reconstruction and sense of humor were predictors of SE. Considering the age and sense of humor and other characteristics of women with BC, this study can help reduce the impact of the disease on their BI and BE and improve their happiness, which has good research significance. This manuscript is suitable for publication in Int. J. Environ. Res. Public Health, but the following modifications are recommended prior to publication.
- There are some formatting errors in the article, please check and modify them.
Response: Thank you very much for pointing this out, we have modified this issue.
- On page 2, line 61, "BC is the only one where mutilation is performed" is not precise, please replace it with an appropriate expression.
Response: Thank you very much for pointing this out, we have modified this section (See line 66).
- Page 2, line 70 "lower survival rates may be increased" is not fluent in language, please modify this.
Response: Thank you for your comment, we have modified this section for better understanding (See line 77).
- In Tables1 and 2, "With reconstruction" and "Without sense of humor" are not very convincing due to the small number of people. It is suggested to investigate more data.
Response: No further sample is available for this research, the sample was chosen randomly.
- Please mark where Table 3 is described in the text.
Response: Thank you very much for pointing this out, we have added this information in the manuscript (See line 248).
- "Age" is mentioned in the keyword, but there is not much discussion about it in the text, please add some discussion about it.
Response: Thank you very much for pointing this out, we have added more information in the manuscript (See lines 307-314).
- The format of references is not uniform. For example, Ref. 6, change “2021, Volume 13, 701–709” to “2021, 13, 701–709”.
Response: Thank you for your comment, we have modified this section for better understanding.
- Introduction, the authors talked about cancer, the latest cancer diagnosis and treatment should be mentioned. In order to support this statement, the following recently published important related papers should be cited: Adv Mater. 2022, 34, 2106388; Sci. China: Chem. 2023, 66, 10.1007/s11426-022-1477-x.
Thank you very much for pointing this out, we have added more information in the manuscript (See lines 48-49).
We hope we have now answered all your comments and we are looking forward to hearing from you again.
Thank you very much,
Sergio Álvarez Pardo, Sport Science Degree

Round 2
Reviewer 1 Report
Dear authors,
Your manuscript, "Factors associated with body image and self-esteem in mastectomized breast cancer survivors", shows the results of the self-perception of breast cancer patients after surgery. I believe this manuscript would support the previous publication of this research group (Alvarez-Pardo et al., 2023).